# The Role of Mass Spectrometry in Hepatocellular Carcinoma Biomarker Discovery

**DOI:** 10.3390/metabo13101059

**Published:** 2023-10-08

**Authors:** Eric Yi-Liang Shen, Mei Ran Abellona U, I. Jane Cox, Simon D. Taylor-Robinson

**Affiliations:** 1Department of Radiation Oncology and Proton Therapy Center, Linkou Chang Gung Memorial Hospital and Chang Gung University, Taoyuan City 333, Taiwan; 2Clinical Metabolomics Core Laboratory, Linkou Chang Gung Memorial Hospital and Chang Gung University, Taoyuan City 333, Taiwan; 3Department of Surgery and Cancer, Faculty of Medicine, Imperial College London, London W2 1NY, UK; 4School of Clinical Medicine, University of Cambridge, Cambridge CB2 0SP, UK; 5The Roger Williams Institute of Hepatology, Foundation for Liver Research, London SE5 9NT, UK; 6Faculty of Life Sciences & Medicine, King’s College London, London SE5 8AF, UK

**Keywords:** mass spectrometry, biomarkers, hepatocellular carcinoma, metabolomics

## Abstract

Hepatocellular carcinoma (HCC) is the main liver malignancy and has a high mortality rate. The discovery of novel biomarkers for early diagnosis, prognosis, and stratification purposes has the potential to alleviate its disease burden. Mass spectrometry (MS) is one of the principal technologies used in metabolomics, with different experimental methods and machine types for different phases of the biomarker discovery process. Here, we review why MS applications are useful for liver cancer, explain the MS technique, and briefly summarise recent findings from metabolomic MS studies on HCC. We also discuss the current challenges and the direction for future research.

## 1. Introduction

Mass spectrometry (MS) and nuclear magnetic resonance (NMR) spectroscopy are the two principal methods used in metabolomics to measure small molecules in biological samples. MS has been widely applied to discover biomarkers for diagnostic, prognostic, and stratification purposes or to better understand the underlying pathological processes of different diseases. Among these, liver cancer, one of the major malignancies with high mortality, has been extensively studied using mass spectrometry of tissue or biofluid samples during the past decade. Here, we review why MS applications are useful for studying liver cancer, provide an overview of the technique, and summarise the MS findings published to date.

## 2. Hepatocellular Carcinoma and the Metabolomic Approach

### 2.1. Epidemiology of HCC

Liver cancer is the sixth most frequently diagnosed malignancy and the third leading cause of cancer-related mortality worldwide. In 2020, liver cancer was responsible for 905,700 new cases and 830,200 cancer deaths globally. The incidence rates of liver cancer are highest in Eastern Asia, South-Eastern Asia, and Northern and Western Africa [1]. Moreover, new cases of liver cancer are expected to increase by 55% in the next 20 years [2].

Hepatocellular carcinoma (HCC) accounts for 85–95% of primary liver cancers [3]. Various risk factors contribute to the development of HCC, including chronic infection with either hepatitis B virus (HBV) or hepatitis C virus (HCV), alcohol consumption, and metabolic disorders, such as obesity and type 2 diabetes. These risk factors, however, show geographical variations. For instance, in regions of high incidence, such as China and Western Africa, chronic HBV infection and aflatoxin are the prevailing risk factors, whereas HCV infection predominates in Japan and Egypt [4]. In contrast, in most developed nations, the leading risk factors for HCC are associated with alcohol consumption, metabolic syndrome, type 2 diabetes, obesity, and metabolic-associated fatty liver disease (MAFLD) [5]. Notably, MAFLD-related HCC constitutes 10–20% of the cases in the USA and Europe [6,7].

### 2.2. Metabolic Reprogramming in HCC

Metabolic reprogramming, proposed as an emerging hallmark of cancer in 2011, has garnered significant attention and interest within the scientific community [8]. This concept is pivotal to the evolving understanding of HCC. As the predominant primary hepatic malignancy, HCC is intricately associated with diverse metabolic alterations. The liver, an essential organ responsible for maintaining physiological homeostasis within the human body, carries out a wide range of functions. Among these are the regulation of blood glucose levels, lipid metabolic pathways, and the enzymatic detoxification of various substances. HCC introduces significant disruptions in these carefully regulated metabolic networks, leading to a range of metabolic changes reflective of the underlying pathophysiology of the malignancy. In the following subsections, specific alterations in glucose and lipid metabolism associated with HCC will be reviewed.

#### 2.2.1. Glucose Metabolism in HCC

In the 1920s, Otto Warburg observed that in certain cancer cells, even when sufficient oxygen was present, there was increased uptake of glucose and the production of lactate [9]. This phenomenon, known as “aerobic glycolysis” or the “Warburg effect,” was initially thought to be a consequence of impaired mitochondrial respiration [10]. However, this altered glucose utilisation in cancer cells was subsequently recognised as part of a rigorously regulated metabolic reprogramming implemented to support tumour growth [11,12].

In HCC, upregulated aerobic glycolysis [13,14] and changes in related enzymes in the Krebs cycle have been reported [15]. The Krebs cycle plays a pivotal role not only in energy generation but also in synthesising various cellular building blocks essential for proliferation. In HCC, citrate synthase, the rate-limiting enzyme that catalyses the formation of citrate from acetyl-CoA and oxaloacetate (OAA), has been found to be upregulated [16,17]. In addition, an elevation in serum citrate levels has been reported in HCC patients compared with healthy individuals [18]. Three oncometabolite-related enzymes, namely succinate dehydrogenase (SDH), fumarate hydratase (FH), and isocitrate dehydrogenase 1/2 (IDH1/2), have been identified as dysregulated. Studies have demonstrated the downregulation of three subunits (SDHA/B/C) of the SDH complex in HCC, along with an increase in succinate levels in both HCC cell lines and tumour tissues. Moreover, the subunits SDHB and SDHC are linked with a subset of tumours correlated with poor prognostic outcomes [15,19,20].

The downregulation of FH expression has been observed in a particularly aggressive subset of HCC patients with portal vein thrombosis (PVT) compared to those without PVT [21]. This may suggest that the dysregulation of FH plays a role in facilitating vascular invasion within HCC. Further analysis by The Cancer Genome Atlas (TCGA) group has identified an HCC subclass characterised by a higher prevalence of IDH1/2 mutations and an associated decline in survival [22]. Moreover, mitochondrial malate dehydrogenase (MDH2), the enzyme responsible for converting malate back into OAA, has been found to be upregulated in HCC tissue, according to the Oncomine datasets [11]. In brief, while the alterations of the Krebs cycle enzymes among various aggressive subclasses of HCCs are not yet fully understood, these changes are tightly connected to glycolysis and tumour prognosis.

#### 2.2.2. Lipid Metabolism in HCC

Lipids, constituting a diverse group of metabolites that are insoluble in water, serve crucial functions in energy storage, the composition of membranes, and signal transduction. The role of lipid metabolism in cancer, particularly in HCC, has become an area of interest. Risk factors for HCC, namely obesity and metabolic-associated steatohepatitis (MASH), have been demonstrated to contribute to hepatocarcinogenesis through mechanisms involving either the gut microbiome or endoplasmic reticulum (ER) stress [23,24]. These risk factors further correlate with insulin resistance and alterations in the metabolism of lipids and fatty acids [25,26]. Within a cohort of HCC predominantly characterised by viral hepatitis, research has also unveiled a connection between enhanced lipogenesis and the development of HCC mediated through the AKT-mTORC1-RPS6 signalling pathway [27].

Fatty acids are central players in lipid metabolism. After entering the cytoplasm, fatty acids undergo activation to form acyl-coenzyme A (CoA), which can then be utilised in the synthesis of phospholipids and triglycerides or regenerated by lipid catabolism [28]. In addition to the uptake of exogenous fatty acids, de novo fatty acid synthesis empowers cancer cells, including those in HCC, to manufacture a diverse spectrum of fatty acids, thereby augmenting their adaptive capacity [29]. Moreover, fatty acid oxidation (FAO) serves as the predominant energy-producing pathway in non-glycolytic tumours. It has been found to be enhanced in cancer cells with high lipogenic activity [30]. It is crucial to recognise that shifts in the metabolic reprogramming of fatty acids are not uniform, varying depending on factors such as the cancer type, specific carcinogenic pathway, and microenvironment.

Additionally, bile acids, essential derivatives of cholesterol synthesis within the liver, play a significant role in lipid metabolism and are intrinsically connected with various signalling receptors and pathways. Among these, the farnesoid X receptor (FXR) is the most extensively studied and has been correlated with HCC. As a nuclear hormone receptor, FXR is prominently expressed in the small intestine, particularly in the ileum, as well as in the liver and kidney [31]. Physiologically, approximately 90 per cent of bile acids are formed in hepatocytes through the classical bile acid synthesis pathway, with cholesterol 7α-hydroxylase (CYP7A1) as the rate-limiting enzyme. Once synthesised, hepatocytes can secrete bile acids into bile canaliculi via the bile salt export pump (BSEP) or other transporters, such as the organic solute transporter (OST) α/β complex, the multidrug resistance-associated protein (MRP) 3 and MRP4. Intriguingly, these transporters fall under the regulatory control of hepatic FXR.

In the inflamed hepatitic liver, the transcription of FXR is downregulated by nuclear factor kappa-B (NF-κB), leading to the downregulation of a small heterodimer partner (SHP). Since SHP inhibits CYP7A1, this downregulation results in a boost in bile acid synthesis. Meanwhile, this downregulation also lowers the activities of FXR-regulated efflux transporters, resulting in the accumulation of hepatic bile acids. Notably, the elevation in specific hydrophobic bile acids such as lithocholic acid (LCA), deoxycholic acid (DCA), and chenodeoxycholic acid (CDCA) can further contribute to hepatocyte damage [32] and hence induce the development of HCC [33,34,35].

### 2.3. The Role of Metabolomics in HCC Biomarker Discovery

Studies have been conducted to identify potential HCC biomarkers; for example, Zhou et al. proposed a plasma microRNA panel specifically for the diagnosis of hepatitis B virus-associated HCC [36]. Additional research has identified proteins such as PIVKA-II (Protein induced by vitamin K absence or antagonist-II) [37,38,39,40], AFP-L3 (lens culinaris agglutinin-reactive fraction of AFP) [41], and osteopontin [42] as potential diagnostic biomarkers. While biomarkers should offer potential clinical applications in areas such as screening, diagnosis, prognosis, patient stratification, and treatment response evaluation, the application of these novel biomarkers for HCC in routine clinical settings is limited. The restricted sensitivity and specificity of known markers often preclude their widespread clinical application, except in a limited number of cases. Notably, those proposed biomarkers are predominantly large molecules—such as RNA sequences and proteins—identified via genomic and proteomic methodologies, rather than small metabolites that can be robustly evaluated through metabolomics.

Metabolomics is an integral component in the study of metabolic reprogramming in cancer, offering a comprehensive analytical approach for the quantification and evaluation of metabolites influenced by either intrinsic or extrinsic factors. Unlike genomics, transcriptomics, and proteomics, metabolomics is the “-omics” field most intimately linked to biological phenotypes [43,44]. Both NMR and MS deliver high-throughput data, encompassing a broad spectrum of metabolites, and are valuable in the identification of these metabolites. The inherent capability of metabolomics to provide a comprehensive analysis of small molecules makes it a potential field for the discovery of biomarkers for HCC.

## 3. Understanding Mass Spectrometry and Its Role in Metabolomics

### 3.1. Principles of Mass Spectrometry

Mass spectrometry (MS) is a principal technique in metabolomics, utilising mass-to-charge (m/z) ratios to isolate and quantify specific metabolites. This method possesses high sensitivity and complements nuclear magnetic resonance (NMR) spectroscopy studies. Typically, MS is integrated with an additional separation modality, such as high-performance liquid chromatography (HPLC) or ultra-performance liquid chromatography (UPLC). Gas chromatography also serves as one of the routine chromatographic interfaces. The chromatographic retention time aids in both the identification and separation of metabolites with identical m/z ratios. Collectively, the integration of HPLC with MS facilitates high-throughput metabolite analysis and identification.

### 3.2. Components of Mass Spectrometry

A mass spectrometer is fundamentally composed of three principal elements: an ionisation source, a mass analyser, and an ion detector. Following chromatographic separation, an ionisation source is employed to convert metabolites of interest into gaseous ionised forms. Various ionisation techniques have been established, with electrospray ionisation (ESI) being among the most widely used. ESI was initially introduced by Dole and subsequently adapted for mass spectrometry applications by Fenn in the 1980s [45]. The technique has garnered widespread adoption owing to its synergistic compatibility with high-performance liquid chromatography (HPLC) [46]. The ionised metabolites are subsequently channelled into the mass analyser via a series of voltage gradients, where they undergo separation based on their mass-to-charge (m/z) ratios. Frequently employed types of mass analysers encompass the quadrupole (Q), time-of-flight (TOF), magnetic sector, and ion trap (IT) configurations. Finally, an ion detector quantifies the relative abundance of ions that successfully pass through the mass analyser.

The primary role of the mass analyser is to separate incoming ions based on their mass-to-charge (m/z) ratios. Since each type of mass analyser has its own set of advantages and disadvantages, it is often the practice to use hybrid mass spectrometers that combine multiple analysers. In a triple-quadrupole mass spectrometer (TQ), three sequential quadrupole mass analysers are interconnected, providing a platform that is widely utilised in various analytical applications. Each quadrupole assembly consists of four parallelly aligned metal cylinders, and both radiofrequency (RF) and direct current are applied to these. By fine-tuning the RF and current settings, the first and third quadrupoles in a TQ, often referred to as MS1 and MS2, respectively, can either permit the passage of all charged ions (when in RF-only or scan mode) or selectively allow ions with a specific m/z range to pass (in selected ion monitoring, or SIM, mode).

The second quadrupole typically functions as a collision cell, facilitating the fragmentation of parent ions chosen by the first quadrupole into daughter ions. Consequently, a TQ is also commonly referred to as a QqQ or MS/MS [47,48]. Notably, the capacity for both MS1 and MS2 to select specific m/z ratios enhances the overall selectivity of the instrument. The quadrupole time-of-flight (Q-TOF or QqTOF) mass spectrometer is another prevalent hybrid instrument. It integrates a quadrupole, a collision cell, and a time-of-flight (TOF) mass analyser. TOF analysers distinguish charged ions based on their transit time, which is directly proportional to the square root of their m/z ratios, offering the advantages of high mass resolution and accuracy [47]. When paired with the quadrupole’s high collision efficiency, Q-TOFs become particularly effective for untargeted metabolic profiling, a topic further discussed in the subsequent subsections.

### 3.3. Mass Spectrometry in Untargeted and Targeted Metabolomics Strategies

Metabolomics studies generally fall into two main categories: untargeted and targeted approaches (Table 1). Untargeted metabolomics records all metabolites detectable by a given method and aims to identify new biomarkers without pre-existing assumptions about their relevance. On the other hand, targeted metabolomics focuses on validating potential biomarkers, either based on the prior literature or findings from untargeted studies, by precisely measuring the concentrations of a selected set of metabolites. Some methods, often described as semi- or pseudo-targeted, incorporate elements from both untargeted and targeted approaches [49].

In untargeted metabolomics, quadrupole time-of-flight (QTOF) mass spectrometers are frequently used for comprehensive metabolic profiling. These instruments inherit the advantages of TOF analysers, delivering high-resolution and accurate mass measurements while covering an extensive m/z range in a short period of time. Furthermore, the preceding quadrupole and collision cell can function in either a broad scan or selective ion filtration mode and either a general scan or fragmentation mode, respectively. In contrast to QTOF systems, TQ mass spectrometers possess lower resolution and are restricted in the number of targets they can concurrently detect. However, they are frequently employed in targeted metabolomics for the quantification of known metabolites. By utilising Selected Reaction Monitoring (SRM) modes, in which both the precursor and product ions are selectively filtered by the first and second quadrupoles (MS1 and MS2), TQ systems achieve a higher level of selectivity. As a result, TQ instruments attain improved signal-to-noise ratios, a feature crucial for precise quantification in targeted metabolomics. Generally, a single injection on a TQ system scans multiple m/z values, thereby facilitating the monitoring of several ions and corresponding reactions. This mode of operation is referred to as Multiple Reaction Monitoring (MRM) [47,48,50].

## 4. The Application of Mass Spectrometry to Discovering Biomarkers for HCC

### 4.1. The Prevalence of Mass Spectrometry

Numerous studies have identified a range of metabolites as potential biomarkers for distinguishing HCC from cirrhosis, chronic hepatitis, or healthy individuals. In a review of the literature previously conducted by our group [51], we analysed a total of 84 studies focused on metabolomic investigations of diagnostic biomarkers for HCC. Notably, the majority (66 out of 84 studies) employed MS as their principal analytical technique. This underscores the pivotal role that mass spectrometry occupies in metabolomic research for HCC. Additionally, of these 84 studies, 54 utilised blood samples and nine analysed urine samples, indicating a preference for minimally invasive methods for biomarker sampling.

### 4.2. HCC Biomarkers across Blood and Urine Samples

Our previous investigation summarized the crucial importance of analytical robustness as an evaluation criterion for metabolic biomarkers distinguishing HCC from control populations [51]. Among blood-derived samples, primary bile acids, namely glycocholic acid, taurocholic acid, and taurochenodeoxycholic acid, exhibited the highest levels of analytical robustness as HCC biomarkers [52,53,54,55,56]. However, further studies are warranted to ascertain whether the levels of these bile acids are indicative of HCC or cirrhosis, as these markers demonstrate various alterations depending on the group compared. Other metabolites, including gluconic acid and hypoxanthine, also showed variable alterations depending on the comparison group. Of particular note, the altered levels of trimethylamine-*N*-oxide (TMAO) and 2-hydroxybutyric acid display high consistency in the alteration in blood samples. TMAO has previously been correlated with microbial dysbiosis [57,58], while 2-hydroxybutyric acid has been linked to both energy metabolism [59] and the gut microbiome [60]. Considering that bile acids also interact with the gut microbiota, this illustrates the potential role of the gut microbiota in the pathophysiology of HCC.

In urine-based biomarker research, a noticeably smaller set of metabolites has been discovered for HCC diagnosis, likely due to the filtration processes occurring in the kidneys. Notably, TMAO levels were reduced in HCC patients, consistent with previous blood-based studies. This consistency across different biological matrices underscores the potential utility of TMAO as a robust biomarker for HCC.

It is relevant to note that 15 studies employed tissue samples, introducing an alternative strategy in biomarker discovery. Conventionally, putative biomarkers are identified through untargeted metabolomics analyses, quantified via targeted LC-MS assays, and subsequently validated in independent external cohorts, as illustrated in Table 1. Nonetheless, metabolite annotation remains a time-consuming bottleneck in untargeted metabolomics research. The employment of tissue samples may accelerate this annotation process. This is potentially due to several factors: enhanced signal intensity of metabolites, an increased number of identifiable metabolites, especially those implicated in interconnected metabolic pathways, and the availability of multi-omics information to support metabolic reprogramming. Further details on this topic will be covered in Section 5.2.

The latest findings from illustrative studies published in 2022 and 2023 are summarised in Table 2. Some of the metabolites selected to be included in predictive models to discriminate HCC from non-HCC control groups in the studies are marked with an asterisk. These data show a similar trend as that observed in our previous review [51] in terms of the type of sample used (blood being the most common), the array of techniques employed, and the diversity of metabolites presented as potential biomarkers. This demonstrates the increasing applicability of MS studies to HCC biomarker discovery. But at the same time, the limited overlap of findings between studies and the paucity of studies that were multi-centred or included a validation cohort (two out of ten) are some of the reasons why it is not yet possible to extrapolate results between published studies to hone in on a set of metabolites to be taken forward for further investigations for clinical application. Further challenges in the field will be discussed in Section 5.1.

### 4.3. Pathway Analysis and the Functional Significance of Reported Candidate Metabolomic Biomarkers

Having identified metabolites that show aberrant levels in HCC, it would be prudent to elucidate the pathophysiology of their alteration, both to better understand the underlying disease process and to evaluate their merit as candidate biomarkers. For example, metabolites in the central energy metabolic pathways of glycolysis and Krebs cycle have been found to be altered, in agreement with the existing understanding of changes in HCC discussed in Section 2.2.1. Pyruvate and lactate are increased while a number of intermediates in the Krebs cycle, such as succinate, fumarate, and malate, are reduced in tumour tissue compared to adjacent non-cancerous tissue [71]. Thus, targeting the rate-limiting steps in the glycolytic pathway could be a potential treatment approach [72].

### 4.4. Integrating Mass Spectrometry Data with Other Omics Approaches

The strategy to survey a wide array of metabolites using untargeted methods followed by targeted assays to discover and confirm differential metabolites described in Section 4.2 and Table 1 is an empirical way to discover potential biomarkers. Alternatively, a deductive approach could be implemented by seeking metabolites informed by pathways found to be altered in other omics. For example, using findings from The Cancer Genome Atlas (TCGA) project [73], Dumenci et al. [74] focused on *Carbamoyl Phosphate Synthetase 1* (*CPS1*), one of the genes found to be most frequently mutated in HCC, which has a metabolic role. A gene–metabolite network was constructed using online databases to inform metabolites that may be affected due to *CPS1* mutation. In reverse, results from metabolomic investigation can be used to inform investigative research as well. After identifying that linoleic acid and phenol were depleted in both portal vein and stool samples from patients, Liu et al. [68] showed that supplementation of these metabolites in HCC cell line suppressed proliferation and induces apoptosis.

Another more comprehensive investigation could be conducted by applying multiple different “-omic” platforms on the same cohort. Zhang et al. [75] performed genomic, transcriptomic, MS-based proteomic, and metabolomic methods, as well as cytometry and single-cell analysis on tissue samples from eight HCC patients. They found that there is substantial heterogeneity in tumour tissues. Intriguingly, it was reported that the metabolome correlated better with the immunome than the transcriptome and proteome.

## 5. Challenges and Future Directions

### 5.1. Challenges and Considerations in MS-Based Biomarker Discovery for HCC

Notwithstanding the significant potential of metabolomics in unearthing novel biomarkers for hepatocellular carcinoma (HCC), several challenges persist, largely owing to the lack of a universally adopted consensus on study quality. Firstly, adherence to critical experimental steps such as sample randomisation and the employment of pooled quality control samples should be rigorously documented in the untargeted analysis phase. Secondly, during the quantification phase, where targeted assays are employed, researchers are responsible for disclosing results from method validation in tandem with quality assurance and quality control outcomes. Thirdly, during data analysis, clinical characteristics that could act as confounding variables must be reported and appropriately accounted for. With respect to metabolite annotation, a minimum confidence level of Category II, confirmed by matching fragmentation data to MS/MS spectral libraries, is essential [76]. Lastly, after quantification via targeted assays, external validation is necessary—a step often neglected, as evidenced by the fact that only a quarter of studies included independent validation cohorts [51]. This final stage is critical for the successful translation of identified biomarkers into clinical practice.

### 5.2. Further Directions—Mass Spectrometry Imaging

Emerging technologies have increasingly been incorporated into MS applications, among which mass spectrometry imaging (MSI) stands out as particularly noteworthy for biomarker discovery. Traditional metabolomic methods, which rely on the analysis of bulk tissue extracts, often sacrifice crucial spatial information about metabolite sub-localisation during the extraction process. This limitation is further compounded by the potential for adjacent background tissue to contaminate tumour samples or for small tumours to contaminate background liver tissue. However, the use of MSI circumvents these issues by generating in situ mass spectra pixel by pixel, thus preserving the spatial integrity of the sample [77].

While tissue-derived biomarkers are often criticised for their invasive collection methods, their utilisation holds unique advantages. Firstly, given their higher metabolite concentrations, HCC tumour samples and their paired normal liver tissues can aid in metabolite annotation. On the other hand, this approach can serve as a confirmatory step for proposed biofluid-based biomarkers and provide a foundation for future multi-omics studies concerning the mechanisms of hepatogenesis and their link to biofluid biomarker variations. Finally, various mass spectrometry imaging ionisation methods are available for tissue samples, each excelling in ionising specific categories of molecules, which can facilitate the metabolite annotation process.

A variety of ionisation methods for MSI have been developed, including secondary ion mass spectrometry (SIMS), desorption electrospray ionisation (DESI), and matrix-assisted laser desorption ionisation (MALDI). SIMS offers ion images with the highest resolution compared to other MSI techniques and is capable of exploring metabolites at deeper tissue layers. However, this “hard” ionisation limits the detection of larger molecular entities. Conversely, both MALDI and DESI utilise “soft” ionisation. MALDI excels in detecting metabolites within a higher mass range (>1000–2000 Da) and offers superior spatial resolution (50–200 µm) compared to DESI (~100 µm) [78,79]. DESI, on the other hand, is particularly adept at lipid analysis, requires minimal sample preparation, and operates efficiently at atmospheric pressure [48,79,80,81].

The existing literature supports the superiority of MSI over conventional tissue extract methods in the analysis of phospholipid levels in HCC. While studies utilising tissue extracts have underscored the discriminative capabilities of specific phospholipid species, namely phosphatidylcholine (PC), phosphatidylethanolamine (PE), and phosphatidylserine (PS) [82], the results have been inconsistent. Some studies reported increased levels of PC and PE in HCC tumour tissue [14], while others reported reduced levels of PE, PS, and phosphatidylinositol (PI) in HCC tissues [83]. These inconsistencies may be partly attributed to patient heterogeneity but also illustrate the inherent limitations of tissue extracts, particularly the contamination between tumour and adjacent non-tumour tissues.

In contrast, MSI studies have produced more consistent results. One study using MALDI found elevated levels of non-ether-linked PEs and phosphatidylinositols (PIs) with specific acyl chains in HCC tumours [84]. Furthermore, a transcriptomic analysis confirmed the increase in PE, showing elevated expression of PE cytidylyltransferase (PCYT2), the rate-limiting enzyme in PE biosynthesis, in HCC compared to normal liver tissues [85].

Although the application of MSI to HCC is still relatively new, its utility is clearly increasing. MALDI-MSI is the most commonly used MSI technique, largely due to its maturity. While most MALDI studies have focused on proteomics, given MALDI’s advantage in detecting larger molecules [86,87], several have explored its applicability in metabolomics. For example, one study revealed increased monounsaturated fatty acid (MUFA)-phosphatidylcholine (PC) and decreased plasmalogens in HCC compared to background alcohol- or NASH-related liver [77]. Another study demonstrated increased metabolic heterogeneity and reprogramming in HCC tumours involving metabolites such as arginine, PC, and fatty acids [88].

With respect to DESI-MSI studies, one DESI-MSI study proposed that triglyceride (TG) 16:0/18:1(9Z)/20:1(11Z) and TG 16:0/18:1(9Z)/18:2(9Z,12Z) was significantly higher and lower in HCC tumour regions than non-tumour regions, respectively [89]. Another DESI variant, Air Flow-Assisted DESI (AFADESI), identified alterations in β-alanine, arginine, and proline metabolism as well as in fatty acid biosynthetic pathways [90].

Overall, MSI offers several benefits over traditional tissue extract methods, notably the preservation of spatial integrity, which enables more precise identification and localisation of tumour-specific phospholipid changes, in addition to minimal sample preparation requirements. As advancements in MSI technologies continue, they are likely to play an increasingly critical role in the discovery and validation of biomarkers for HCC, thereby enriching the fields of diagnosis, prognosis, and treatment monitoring.

### 5.3. Conclusions

Mass spectrometry is a promising technology with the potential to discover biomarkers for various purposes in the natural history of HCC, including diagnosis, prognosis, and disease stratification. It has been utilised in numerous studies, and many candidate biomarkers have been reported, with little consistency to date. Challenges remain to be overcome: in terms of study design, the sample sizes of many studies are small, with differing underlying aetiologies, tumour morphologies, and tumour stages; in terms of methodology, better experimental rigour, metabolite annotation, and the need for external validation are necessary to harness the power of MS fully. Ideally, multi-centre studies with larger study numbers should be employed with more uniform tumour categorisation, sample collection, and scientific protocols in order to provide more definitive answers on biomarker feasibility in this arena.

## Figures and Tables

**Table 1 metabolites-13-01059-t001:** An overview of the phases of biomarker discovery using mass spectrometry.

Phase	1. Discovery	2. Quantification	3. Validation
Goal	To identify differential metabolites without prior assumption	To accurately quantify the levels of differential metabolites	To confirm differential metabolites using independent cohorts
Method used	Untargeted	Targeted	Targeted
Mass spectrometer of choice	Quadruple time of flight	Triple quadruple	Triple quadruple

**Table 2 metabolites-13-01059-t002:** Summary of illustrative mass spectrometry-based metabolomic findings in hepatocellular carcinoma from studies published in 2022 or 2023.

Publication	*n*	Sample	Technique	Main MS Findings in HCC
Li et al. [61]	HCC: 200 CHB: 200	Plasma	LC-MS	Phosphatidylcholines significantly downregulated
Li et al. [62]	HCC: 68 LC: 33 HC: 34	Serum	LC-MS	Alterations of the levels of five metabolites: taurochenodeoxycholic acid, glycochenodeoxycholate, ouabain, theophylline, and xanthine
Liu et al. [63]	HCC: 104 LC: 76 HC: 10	Plasma	GC-MS	Increased: trans–trans-muconic acid and oxoglutaric acid * Decreased: montanic acid, oleamide, triethylene glycol, 2-picolinic acid, heptaethylene glycol *, *N*-formylglycine *, citrulline *, and 4-(dimethylamino)azobenzene
Fan et al. [64]	HCC: 43 HC: 47	Urine	APGD-MS	Increased: acetic acid, creatine, propionic acid, glycolic acid, cyanoacetic acid, nicotinic acid, heptenoic acid, *L*-pyroglutamic acid, *L*-ornithine, perillic acid, and *N*-acetyltaurine
Yue et al. [65]	Discovery: HCC+T2D: 19 T2D: 32 Test: HCC+T2D: 64 T2D: 96 HC: 94	Serum	LC-MS/MS	Increased: 8,15-dihydroxy-5,9,11,13-eicosatetraenoic acid (8,15-DiHETE), hexadecanedioic acid (HDA) *, 15-keto-13,14-dihydroprostaglandin A2 (DHK-PGA2) *, and octadecanedioic acid
Morine et al. [66]	HCC: 20	Tissue and serum	CE-MS	Tissue: increased leucine, valine, tryptophan, isoleucine, methionine, lysine, and phenylalanine Serum: increased leucine, valine, and tryptophan
Qu et al. [67]	HCC: 57 HC: 76	serum	SALDI-MS	A total of 14 lipids containing different lipid types (TAG, CE, PC) were selected as potential lipidomic biomarkers
Liu et al. [68]	Discovery: HCC: 52 HC: 59 Validation: HCC: 50 HC: 50	Serum (portal vein and central), tissue, and stool	LC-MS	Tissue and portal vein serum: increased *DL*-3-phenyllactic acid, *L*-tryptophan, glycocholic acid, and 1-methylnicotinamide; Portal vein and stool: decreased linoleic acid and phenol
Wu et al. [69]	HCC: 93 CHB: 136	Serum	LC-MS/MS	Increased: phenylalanine, tyrosine ratio, and the kynurenine-to-tryptophan ratio Decreased: leucine, lysine, threonine, tryptophan, valine, serotonin, taurine, and tryptophan ratio, BCAA/aromatic amino acids ratio, BCAAs/tyrosine ratio, Fischer’s ratio, and serotonin-to-tryptophan ratio
Pan et al. [70]	HCC: 30 LH: 29 CHB: 30	Serum	LC-MS	Increased: taurodeoxycholic acid * and 1,2-diacyl-3-β-d-galactosyl-sn-glycerol * Decreased: 5-hydroxy-6*E*,8*Z*,11*Z*,14*Z*,17*Z*-eicosapentaenoic acid and glycyrrhizic acid *

* Selected to be included in the statistical model in the study to discriminate HCC from non-HCC. APGD-MS: Atmospheric pressure glow discharge mass spectrometry; BCAA: branched-chain amino acids; CE: cholesteryl; CE-MS: capillary electrophoresis–mass spectrometry; CHB: chronic hepatitis B carrier; GC-MS: gas chromatography–mass spectrometry; HC: healthy control; HCC hepatocellular carcinoma; LC: liver cirrhosis; LC-MS: liquid chromatography–mass spectrometry; PC: phosphatidylcholine; and TAG: triacylglyceride.

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
