# Peer review of "The Role of Mass Spectrometry in Hepatocellular Carcinoma Biomarker Discovery"

_metabolites, 2023, doi:10.3390/metabo13101059_

Round 1

Reviewer 1 Report

This paper provides a valuable and informative overview of the role of mass spectrometry in hepatocellular carcinoma biomarker discovery. It covers the principles, methods, applications, and challenges of mass spectrometry in metabolomics, and discusses the current and emerging findings in this area.

The paper is well-written, well-structured, and well-referenced. It is useful for researchers and clinicians who are interested in the field of mass spectrometry and hepatocellular carcinoma.

However, the paper is too redundant in some parts, especially in the introduction (Section 3) and discussion sections. The authors should consider reducing the length of these sections and avoiding repetition of information that has already been presented elsewhere.

The paper also has some specific issues that need to be addressed, such as:

    - In this paper, what is the most important and novel statement by the authors? The authors should clearly state their main contribution and novelty in the abstract and introduction, and highlight how their review differs from previous reviews on the same topic.

    - In this paper, what is most useful for mass spectrometric methods for HCC biomarker analysis? The authors should provide more details and comparisons of the different mass spectrometric methods for HCC biomarker analysis, such as their advantages, disadvantages, limitations, and applications. They should also explain why they chose to focus on electrospray ionisation (ESI), quadrupole time-of-flight (QTOF), and triple quadrupole (TQ) methods, and how they relate to other methods such as matrix-assisted laser desorption/ionisation (MALDI) or Fourier transform ion cyclotron resonance (FTICR).

    - In this paper, what specimens are useful for HCC biomarker analysis? The authors should discuss the pros and cons of using different specimens for HCC biomarker analysis, such as blood, urine, tissue, bile, saliva, etc. They should also mention the challenges and solutions for sample collection, storage, preparation, and analysis of these specimens.

    - In this paper, what metabolites are useful for HCC diagnosis? The authors should provide more evidence and references to support their selection of metabolites that are useful for HCC diagnosis, such as primary bile acids, TMAO, pyruvate, and lactate. They should also discuss the biological significance and mechanisms of these metabolites in HCC pathogenesis and progression.

    - In this paper, are the validity and reproducibility in HCC biomarker analysis discussed? The authors should elaborate on how to assess and improve the validity and reproducibility of metabolomic biomarkers for HCC diagnosis and prognosis. They should also address the sources of variability and bias that may affect the metabolomic results, such as analytical platforms, laboratories, populations, confounding factors, etc.

    - In this paper, are the multicentered clinical studies cited? The authors should cite more multicentered clinical studies that have used metabolomic approaches to identify HCC biomarkers in large and diverse cohorts. They should also discuss the challenges and opportunities of conducting multicentered clinical studies in this field.

    - In this paper, are the spatial omics cited as well? The authors should explain what spatial omics is and how it can be integrated with metabolomics to provide spatial information and omics data on HCC tissues. They should also discuss the advantages, disadvantages, applications, and future directions of spatial omics in HCC biomarker discovery.

Reviewer 2 Report

The authors have presented the mass spectrometry as one of the important techniques used in HCC metabolomics studies. The topic is interesting but the manuscript needs an extensive revision. My comments are:

1.      Certain lines are repeated (for example, lines 43-44 and 48-49, Warburg effect…). Please rewrite.

2.      The only presented table (Table 1) contains only a few broad observations and no noteworthy data or outcomes related to the review. My suggestion is to change this table to one that includes summary studies on this subject that were conducted using the MS technique (studies on biomarkers for HCC with the methodology utilized)

3.      Section 3.2. about MS components is too long and it only contains general information that is already widely known. This has to be shortened and modified to fit the title.

4.      In the context of HCC metabolomics, the MALDI technique is discussed (again generally) but not described. This section needs to be included since it is a crucial technique for glycan profiling in the identification of HCC biomarkers.

5.      Please update the literature. New studies, chapters, and reviews on this subject can be found using only a Google search.

Moderate editing of English language required

Reviewer 3 Report

Manuscript proposed by Yi-Liang Shen and co-workers (metabolites-2614853) entitled “The role of mass spectrometry in hepatocellular carcinoma biomarker discovery is a review article presenting the application of mass spectrometry in the analysis of liver cancer, provide an overview of the principles behind and summarize the findings from the studies published so far. In my opinion, this is an interesting and well-written article however, needs some changes.      

My comments are presented below.

Major concerns: - more information about the proteome changes in HCC should be presented, - the literature data should be enriched in some new positions presented in databases regarding application of mass spectrometry in such kind of studies, - the sensitivity, matrix effects and other factors that affect the metabolomic and proteomics studies should be discussed, - some things about the derivatization procedure of sample for their better analysis by MS should be presented, - quantitative analysis by MS should be better presented, including application of isotopically labeled standards   Make changes in the text. Check and correct English

This work is well-written and presented. 

Round 2

Reviewer 2 Report

The authors have corrected the manuscript according to the comments.

However, please check throughout the manuscript the compound names: N-formylglycine, N should be italic; L-pyroglutamic acid should be L-pyroglutamic acid;  E and Z should be italic; etc....

Moderate editing of English language required.

Author Response

Answer: We have amended as suggested. The manuscript has been reviewed by a primary English speaker, and amendments have been made where necessary.